# A Loss Curvature Perspective on Training Instability in Deep Learning

**Justin Gilmer** *          **Behrooz Ghorbani** *          **Ankush Garg**          **Sneha Kudugunta**

**Behnam Neyshabur**    **David Cardoze**    **George E. Dahl**    **Zack Nado**    **Orhan Firat**

## Abstract

In this work, we study the evolution of the loss Hessian across many classification tasks in order to understand the effect the curvature of the loss has on the training dynamics. Whereas prior work has focused on how different learning rates affect the loss Hessian observed during training, we also analyze the effects of model initialization, architectural choices, and common training heuristics such as gradient clipping and learning rate warmup. Our results demonstrate that successful model and hyperparameter choices allow the early optimization trajectory to either avoid— or navigate out of—regions of high curvature and into flatter regions that tolerate a higher learning rate. Our results suggest a unifying perspective on how disparate mitigation strategies for training instability ultimately address the same underlying failure mode of neural network optimization, namely poor conditioning. Inspired by the conditioning perspective, we show that learning rate warmup can improve training stability just as much as batch normalization, layer normalization, MetaInit, GradInit, and Fixup initialization.

## 1 Introduction

Optimization of neural networks can easily fail. While recent architectural advances such as skip connections (He et al., 2016a) and Batch Normalization (Ioffe and Szegedy, 2015) have been applied successfully to produce architectures and hyperparameters that reliably train well, even small changes to a trainable configuration can easily result in training that diverges. More generally, producing a configuration that strikes the right balance between stable training and rapid optimization progress on a new domain can be difficult—practitioners and researchers have few reliable heuristics to guide them through the process. As a result, the specific hyperparameter tuning protocol has an outsized influence on the results (Choi et al., 2019; Sivaprasad et al., 2020) and successes often rely on large hyperparameter searches (Nado et al., 2021). Developing a principled understanding of what makes general architectures trainable would allow researchers to more reliably navigate this process and has the potential to dramatically accelerate research into finding better, more scalable architectures.

The focus of the empirical investigation of this work is to better understand what limits the maximum trainable learning rate for deep learning models trained with the typical minibatch stochastic gradient descent (SGD) family algorithms. As part of this investigation, we examine several methods developed by the deep learning community that have enabled training at larger learning rates and improved performance. Many methods have been developed that can achieve this goal, notably normalization, learning rate warmup, gradient clipping (Pascanu et al., 2013), and better model initializations such as Fixup (Zhang et al., 2019b), MetaInit (Dauphin and Schoenholz, 2019), and GradInit (Zhu et al., 2021). While these methods are certainly not exactly equivalent, a key property they all have in common is that they can enable training at larger learning rates when applied to certain models (see for example Figure 1).

---

*Equal Contribution. Correspondence to {gilmer, ghorbani}@google.com.

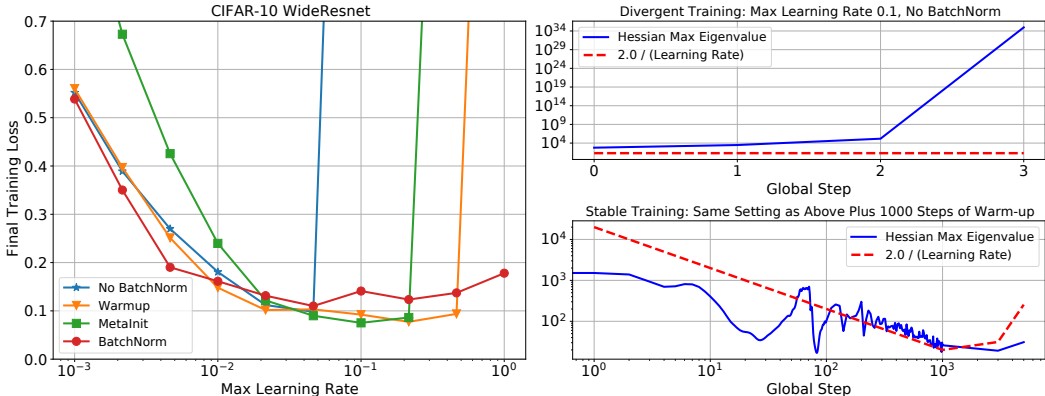

Figure 1: Left: Three different methods which when applied to a WideResnet 28-10 architecture (w/o Batch Normalization) enable training at larger learning rates: learning rate warmup, MetaInit, and adding normalization layers. Each point reports the final training loss after training with cosine decay for 300 epochs. The test performance of the models closely matches the training loss behavior (see Figure 7 of the appendix). Right: The evolution of the largest eigenvalue of the Hessian throughout the training for the No-BatchNorm model with and without warm-up.

A natural hypothesis is that methods which enable training at larger learning rates do so through reducing the *sharpness*[1] of the loss surface during training. Indeed, this hypothesis has already been proposed as one of the beneficial effects of Batch Normalization (Ghorbani et al., 2019; Santurkar et al., 2018) and residual connections (Li et al., 2017), and quadratic models of the loss surface predict that optimization with SGD is unstable when $\lambda_1 > 2/\eta$ (Wu et al., 2018). However, recent empirical investigations into the relevance of quadratic stability bounds to neural network training have either focused on smaller models, focused on full batch training at small learning rates, and do not investigate connections between sharpness, model initialization and learning rate warmup (Cohen et al., 2021; Jastrzebski et al., 2020).

In this work, we design a series of large scale experiments studying the evolution of the loss sharpness as we vary the learning rate, warmup period, initialization, and architectural choices. Our results demonstrate the central role that $\lambda_1$ plays in neural network optimization—maintaining sufficiently small $\lambda_1$ during optimization is a necessary condition for successful training at large learning rates. Consequently, reducing $\lambda_1$ is a primary benefit of proper tuning of a number of architecture and optimization hyperparameters: including model initialization, location of normalization, and warmup schedule. Specifically, we show the following:

- We provide large scale empirical confirmation that training of neural networks with SGD+momentum is stable only when the optimization trajectory primarily resides in a region of parameter space where $\lambda_1 \lesssim 2/\eta$, where $\eta$ denotes the learning rate. This corroborates the theoretical predictions of Wu et al. (2018) and recent empirical observations of Jastrzebski et al. (2020) and Cohen et al. (2021).

- We demonstrate that several successful initialization strategies for architectures without normalization operate primarily by reducing curvature early in training, enabling training at larger learning rates.

- We show that learning rate warmup gradually reduces $\lambda_1$ during training, offering similar benefits to better model initialization. We connect the mechanism by which warmup operates to the dynamical stability model of Wu et al. (2018).

- We show that learning rate warmup is a simple yet competitive baseline for research into better model initialization. We demonstrate that key progress in this area (Dauphin and Schoenholz, 2019; Zhang et al., 2019b; Zhu et al., 2021) can be matched by the application of learning rate warmup and/or gradient clipping alone.

---

[1]Throughout this work we will use the term sharpness to refer to the maximum eigenvalue of the loss Hessian, denoted as $\lambda_1$. See Appendix B for more details.

- Finally, we show that large loss curvature can result in poor scaling at large batch sizes and interventions designed to improve loss conditioning can drastically improve the model's ability to leverage data parallelism.

## 2 RELATED WORK

**Understanding BatchNorm** The loss Hessian has been a central object of study for understanding optimization of neural networks. Santurkar et al. (2018) argues that an important benefit of Batch Normalization is improved smoothness of the loss surface, while Lewkowycz et al. (2020) notes that this is improved smoothness is only observed when higher learning rates are used in combination with Batch Normalization. Our results are generally consistent with this current understanding of Batch Normalization, however some of our experiments provide additional nuance—notably we observe several instances where models suffer from training instability (and high loss curvature) early in training despite using Batch Normalization (see Section 4).

**Evolution of the loss Hessian** Recent research has closely studied the interaction between sharpness and learning rate. Wu et al. (2018) provides a dynamical stability model which predicts that the loss curvature *at convergence* must satisfy[2] $\lambda_1 \leq 2/\eta$. Recent work has provided empirical evidence that $\lambda_1 \lesssim 2/\eta$ often holds well before convergence (Cohen et al., 2021; Jastrzebski et al., 2020). Cohen et al. (2021) focused on full batch training at small learning rates, and observed "progressive sharpening", where $\lambda_1$ increases during training until $\lambda_1 \approx 2.0/\eta$. We observe the progressive sharpening phenomenon also occurs for many models trained with SGD, though we do not investigate batch sizes $\leq 8$, where Cohen et al. (2021) argue that progressive sharpening does not occur. We note that Wu et al. (2018) equation 8 predicts the "edge of stability" is dependent on the batch size and that at small batch sizes this can be will below the $2/\eta$ bound. We confirm this prediction holds even early in training (see Appendix Figure 16). Lewkowycz et al. (2020) proves that for single hidden layer neural networks initialized at point with $\lambda_1 > 2.0/\eta$ and trained with an MSE loss may enter a "catapult" regime—where the loss increases early until a flatter region of the loss surface is found, with divergence occurring in cases where $\lambda_1$ greatly exceeds $2.0/\eta$. In contrast to the simplified setting considered in Lewkowycz et al. (2020), we find that divergence may occur even though $\lambda_1 \ll 2/\eta$ at initialization.

## 3 EXPERIMENTAL SETUP

We investigate models trained on several benchmarks: CIFAR-10 (Krizhevsky, 2009) and ImageNet (Russakovsky et al., 2015) for image classification, LM1B (Chelba et al., 2013) for Language Modeling, and WMT for Neural Machine Translation (NMT). On CIFAR-10 we consider the WideResnet (Zagoruyko and Komodakis, 2016) and DenseNet (Huang et al., 2017) architectures, both with and without Batch Normalization. We consider two variants of the DenseNet architecture. The standard variant from the open sourced code of Zhu et al. (2021) is considered in Figure 5 and Table 1. A less stable variant changes the strides in the average pooling layers to (1,1) is used for Figure 2 and is denoted as Stride-(1,1) DenseNet (see Appendix D.1 for a more detailed discussion). When training without Batch Normalization we consider several initialization strategies including the default "LeCun Normal" initialization, and running MetaInit. As a way to artificially induce worse initializations, we also consider experiments where we scale every variable produced by the default initialization by a constant factor $\alpha$. The NMT models are trained on the WMT'16 EN-DE training set, tuned for hyper-parameters on the WMT'16 EN-DE validation set and evaluated on the WMT'14 EN-DE test set for BLEU scores. For NMT and LM1B Language Modeling, we train 6 layer Transformer models (Vaswani et al., 2017). Inspired from Xiong et al. (2020), we experiment with three Layer Norm settings: pre-Layer Norm, post-Layer Norm (Liu et al., 2020) and no Layer Norm for the transformer models.

Each model is trained with various learning rates using cosine decay (unless mentioned explicitly). For warmup experiments we use linear warmup which starts at 0 and scales linearly to a max value $\eta$ before applying cosine decay. To measure the max eigenvalue of the loss Hessian we use the

---

[2]This is a simplified, potentially loose bound. See the original work for a more general bound that depends on both the loss curvature and the noise covariance matrix.

Lanczos method where the number of iterations varied as needed depending on the architecture (details provided in the appendix).

## 4  EARLY TRAINING INSTABILITY AND THE LOSS HESSIAN

In Figure 2 we plot the curvature at initialization and during training for a series of models trained on different datasets (plots showing final performance for all models can be found in the appendix). Each row indicates a different base model, the left column plots the curvature of the model at initialization and indicates with an 'X' whether or not the model diverges when trained without warmup. On the right we plot the measured curvature and learning rate at a specified point during training. We observe across all datasets that successful training occurs only when optimization enters a region of parameter space where $\lambda_1 \leq 2/\eta$, and that divergent models are outside this region shortly before divergence. At initialization, some models can be successfully trained even when they start out in the unstable region and generally speaking, divergence is more likely for models deeper in the unstable region.

For CIFAR-10 WideResnet, removing batch norm results in a model with higher curvature at initialization and results in divergent models when trained with a learning rate $\eta > .1$. Scaling the WideResnet initialization up by a factor of $1.5$ exacerbates the problem, resulting in even higher curvature at initialization and divergence when $\eta > 10^{-2}$. MetaInit starts the model out at a point with very small $\lambda_1$, and allows training without Batch Normalization at higher learning rates than the default initialization. We also observed that higher learning rates can be unlocked when the models are trained with learning rate warmup. Warmup was particularly effective for models which exhibit large $\lambda_1$ either at initialization or early in training. Other models such as the post activation Resnet-50, and WideResnet w/ Batch Normalization did not benefit from warmup at the considered learning rates (see Appendix).

For the Stride-(1,1) DenseNet experiments, it is noteworthy that the models with Batch Normalization actually start out with higher curvature than the non-BN variants. This is contrary to the generally accepted narrative that Batch Normalization improves the smoothness of the loss surface (Ghorbani et al., 2019; Santurkar et al., 2018). We found that the Batch Normalization models were more unstable than the non-BN variants here, as some models diverged at smaller learning rates. However, when combined with warmup the BN models were trainable at learning rates $\eta > .1$, whereas this did not hold for the non-BN variants, which diverge both with and without warmup at these learning rates. This result suggests that BN still offers training stability for this model, and flatter curvature mid training *if trained with warmup and a higher learning rate*, however no smoothness benefits are observed at initialization. See Appendix D.1 for more details on this phenomenon.

For Resnet-50 trained on ImageNet we compare two different residual blocks: the preactivation block (He et al., 2016b) and the more commonly used post activation block (He et al., 2016a). For the preactivation block, we also consider flipping the order of the ReLU activation and batch normalization, as was considered in Brock et al. (2021). We find that both preactivation models start out in a region of higher curvature relative to the post activation variant, and that these models diverge when $\eta > .5$ whereas the post activation variant is trainable with learning rates as large at 10.

Notably, there are several models in our experiments which diverge despite starting out in a region where $\lambda_1 < 2/\eta$. This occurs for both the pre and post layernorm transformer, and the WideResnet model initialized with MetaInit. We found for these divergent models that the curvature rapidly increases in the initial steps of training, which is partially visible in the mid training plot where we plot the final observed curvature before divergence. Full training curves for these models can be found in the appendix. This implies that measuring $\lambda_1$ at initialization is not always sufficient to predict whether or not the model will be easily trained. Currently, some architectural innovations are motivated by an analysis of either gradient statistics or smoothness at initialization (Liu et al., 2020)—a more robust analysis would consider the evolution of these statistics under SGD.

## 5  THE INTERACTION BETWEEN LEARNING RATE WARMUP, INITIALIZATION AND CURVATURE

The success of learning rate warmup is inconsistent with conventional optimization wisdom, which traditionally suggests adapting the step size to the curvature (see for example the discussion around

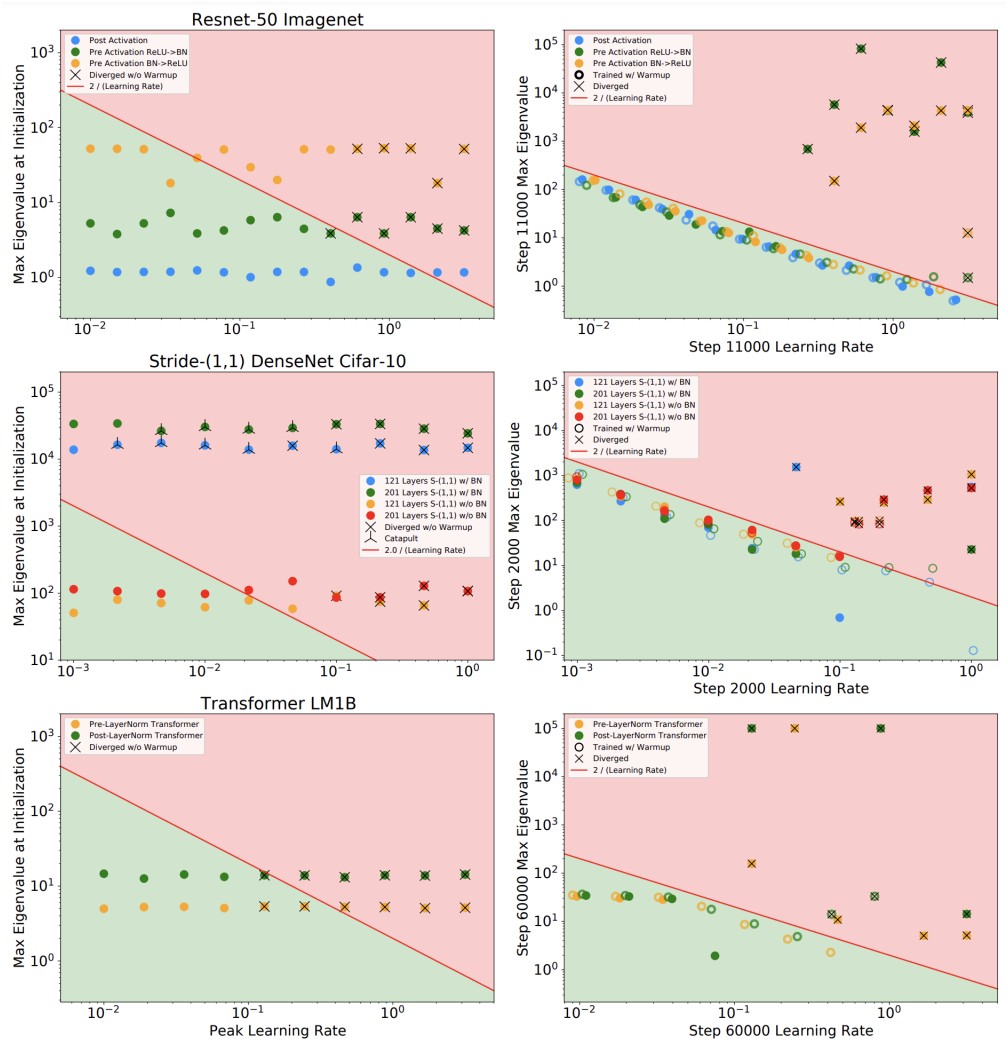

Figure 2: Measurements of the Hessian max eigenvalue of different models at initialization (left) and during training (right). Divergent models are indicated with an X. On the left, we plot $\lambda_1$ at initialization along with the peak learning used during training. On the right, we plot $\eta$ and $\lambda_1$ at a specified step in training. For divergent models on the right plot, we record the last learning rate and max eigenvalue that occur before divergence is detected (defined as observing a NaN in the loss). Across all datasets and models, successful training occurs only when optimization enters a stable region of parameter space where $\lambda_1 \leq 2/\eta$. Models with higher initial loss curvature tend to diverge at smaller learning rates relative to models initialized with flatter curvature, however larger learning rates are possible in these cases when warmup is used. Note, to avoid overlapping points we have applied a small deterministic shift to the x-position of points on the right hand plots. In the case of Stride-(1,1) DenseNets w/ BN we observed some models exhibit catapult behavior (loss increases early followed by normal training), these have been marked accordingly.

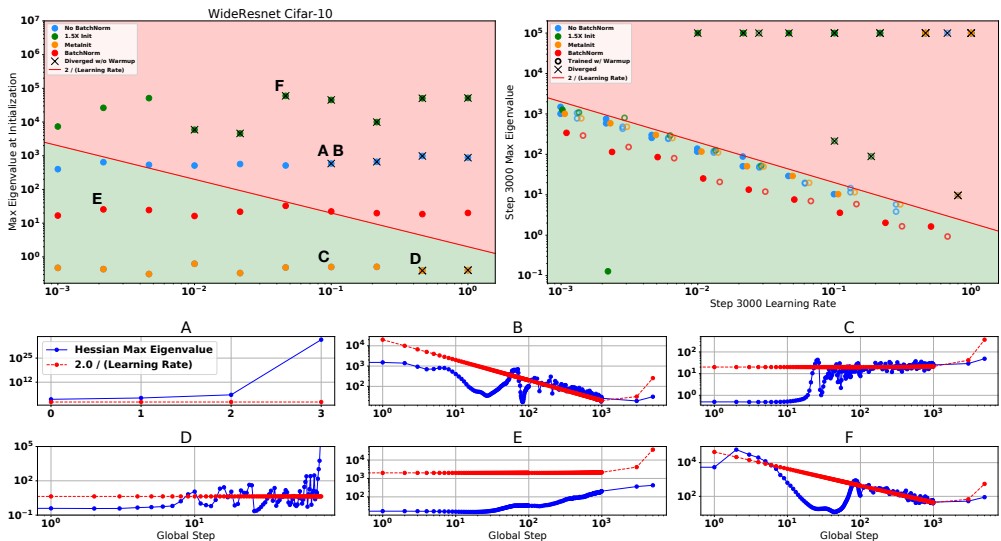

Figure 3: **Top row:** Measured $\lambda_1$ at initialization (left) and mid training (right) for variants of the WideResnet 28-10 model on Cifar10. Both BatchNorm and MetaInit reduce $\lambda_1$ at initialization. **Bottom Row:** Full time evolution of $\lambda_1$ and $\eta$ for select runs (labeled with letters in the top row). **A:** The non-BN variant diverges when trained at $\eta = .1$. **B:** Same peak $\eta$ as (A), but when warmup is used $\lambda_1$ gradually decreases without diverging. **C:** MetaInit allows training without warmup at $\eta = .1$ by starting $\lambda_1$ small. **D:** An example of a model diverging despite starting out in the "stable" region. **E:** An example of progressive sharpening when training with small $\eta$. **F:** Learning rate warmup recovers from an even poorer initialization.

equation 2.4 in McCandlish et al. (2018)). However, with the understanding that $\lambda_1$ is a dynamic quantity whose evolution is tightly coupled with the learning rate schedule, the benefits of a warmup period are more easily understood. We argue that the success of learning rate warmup follows naturally from two properties of training deep models:

1. Models diverge when the learning rate is too large relative to the $2/\lambda_1$ bound.

2. When the learning rate only slightly exceeds $2/\lambda_1$ optimization is unstable until the parameters move to a region with smaller $\lambda_1$ (Wu et al., 2018; Lewkowycz et al., 2020).

The first criteria implies that we can't start $\eta$ off at too large of a value relative to $\lambda_1$ at initialization. The second criteria implies that gradually increasing $\eta$ can gradually "push" the parameters to a region of parameter space where optimization is stable (with lower values of $\lambda_1$). In Figure 4 there is clear evidence for this "pushing", as during the warmup period the we see that $\lambda_1 \approx 2.0/\eta$ holds for a large part of the warmup phase. Furthermore, this approximation holds even as we vary the length of the warmup period. Other examples can be seen in Figure 3 (B and F), and Figure 15 in the appendix.

Warmup is not the only method capable of reducing $\lambda_1$ during training, one can instead initialize the model in a region where $\lambda_1$ starts off small. Consider for example, the points A, B and C in Figure 3. Each point shows optimization of a non-BN WideResnet with peak learning rate of .1. In (A) we see the model diverges within 3 steps without warmup using the default initialization. In (B) we see that a linear warmup period results in $\lambda_1$ progressively decreasing until the peak step size of .1 is reached at step 1000, with no divergence occurring. Finally in (C) we initialize the same model with MetaInit, at which point $\lambda_1$ is small at initialization, and the model can be trained at $\eta = .1$ without warmup.

Similar to the aforementioned MetaInit, the success of related initialization strategies can be explained by reduced $\lambda_1$ early in training. In Figure 5 (left) we look at the evolution of $\lambda_1$ during the GradInit meta optimization process and compare this with simply training the same model using gradient clipping[3]. Both methods result in $\lambda_1$ decreasing dramatically, after which $\lambda_1$ hovers around $2/\eta$.

---

[3]Similar to warmup, gradient clipping reduces the step size in regions of large curvature.

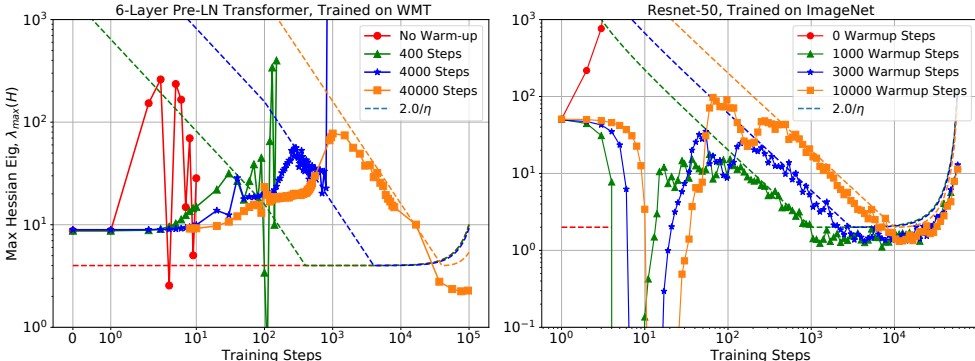

Figure 4: **Learning rate warmup "pushes" the optimization trajectory towards regions of flatter curvature.** Solid lines correspond to the maximum eigenvalue of the Hessian throughout the training. Dashed lines correspond to $2/\eta$. During the warmup period, $\lambda_1$ fluctuates close to the $2/\eta$ bound. The WMT models diverge for 0, 400 and 4000 warmup steps. The pre-activation Resnet-50 model diverges quickly without warmup.

Table 1: **Learning rate warmup can match the performance of recent advances in initialization research.** The non-warmup results were all taken from their respective original works, MetaInit from Dauphin and Schoenholz (2019), GradInit from Zhu et al. (2021) and Fixup from Zhang et al. (2019b). We found in all cases that warmup and gradient clipping could be leveraged to recover from bad initializations, offering similar benefits to these better initialization strategies. The DenseNet-100 Kaiming-Clip 1.0 result is taken directly from Zhu et al. (2021), we found using the same codebase that changing the gradient clipping to 6.0 was enough to closely match the GradInit results. For the 6-layer Transformer w/o LayerNorm we found the default initialization to exhibit extreme curvature, with initial gradient norm on the order of $10^{20}$. We found that warmup was insufficient in this case for any training to occur at this point, however scaling the default initialization down by a factor 4 was enough to recover the performance the normalized model.

| Model | Dataset | Method | $Acc$ |
|---|---|---|---|
| DenseNet-100 | CIFAR-10 | Kaiming-Clip 1.0 | 93.97 |
| DenseNet-100 | CIFAR-10 | GradInit | 94.85 |
| DenseNet-100 | CIFAR-10 | Kaiming-Clip 6.0 (ours) | 94.65 |
| WideResnet 28-10 (w/o BN) | CIFAR-10 | Warmup 1000 | 97.2 |
| WideResnet 28-10 (w/o BN) | CIFAR-10 | MetaInit | 97.1 |
| Resnet-50 (w/o BN) | ImageNet | Fixup | 76.0 |
| Resnet-50 (w/o BN) | ImageNet | MetaInit | 76.0 |
| Resnet-50 (w/o BN) | ImageNet | GradInit | 76.2 |
| Resnet-50 (w/o BN) | ImageNet | Warmup 1000 (ours) | 76.2 |
| Transformer 6L (w/o LN) | WMT | Warmup + Clip + .25x Init (ours) | 27.10 (BLEU) |
| Transformer 6L (w/ LN) | WMT | LayerNorm | 27.01 (BLEU) |

Notably, GradInit starts regular training off at $\lambda_1$ significantly below the $2/\eta$ bound, however the curvature quickly increases within a few steps. Given that initialization and warmup serve similar roles in reducing $\lambda_1$, we expect to be able to achieve similar performance using the two methods. As shown in Table 1 we can easily match key advances in this field by applying learning rate warmup alone (see Appendix for experimental details).

Beyond controlling $\lambda_1$ mid-training, the learning rate $\eta$ controls more general conditioning measures of the loss surface. For example, in Figure 5 we observe that even the MetaInit *gradient quotient*— the conditioning measure directly optimized by this initialization strategy—is controlled by $\eta$ mid training. This again provides further evidence that the primary benefit of this initialization method is to reduce $\lambda_1$ at initialization. As shown, any gains by optimizing the more general gradient quotient must be short lived as the initialization has no control over the long term value.

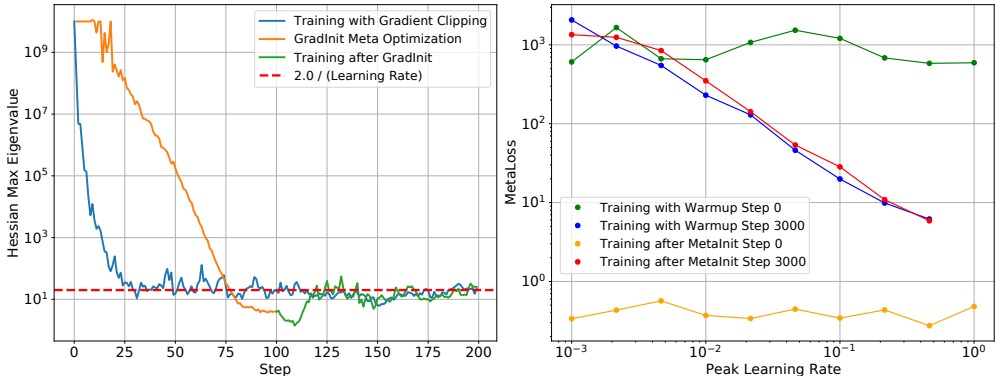

Figure 5: **The mid-training conditioning is determined by the learning rate, not on the initialization method used.** *(a):* We plot the Hessian max eigenvalue during training for two models, a DenseNet model trained with learning rate warmup and gradient clipping, and the same model initialized with GradInit. We also plot the Hessian max eigenvalue during the GradInit meta optimization process. Both GradInit and training with warmup are able to reduce the large curvature of the model. The GradInit algorithm initializes SGD in a flatter region, however this effect only last for a few steps of SGD, at which point the curvature of both the GradInit model and the model trained with warmup are nearly identical. *(b):* We plot the MetaInit MetaLoss during training for two groups of models, ones initialized with MetaInit and ones trained with learning rate warmup. Although the two groups start training with dramatically different values of the MetaLoss, after 3000 steps of SGD the MetaLoss of the two groups of models is almost completely determined by the learning rate used in training, not on the model initialization.

## 6 THE EFFECTS OF CURVATURE ON BATCH SIZE SCALING

So far, we discussed how large loss curvature limits the range of stable learning rates for training. In this section, we highlight how these limits on usable learning rates affect the model's ability to effectively leverage larger batch sizes. Previous research has studied the interplay of the loss curvature and batch size scaling from various different perspectives. Most notably, Shallue et al. (2018) observe that increasing the batch size yields consistent improvements in training speed until a (problem-dependent) critical batch size is reached; increasing the batch size beyond this threshold yields diminishing improvements in training speed. Zhang et al. (2019a) observe that a simple Noisy Quadratic Model (NQM) is able to capture some of the empirical behavior observed in Shallue et al. (2018). Similarly, McCandlish et al. (2018) use quadratic approximations to the loss to provide a closed form expression for the critical batch size as a function of the loss Hessian and the covariance of the stochastic gradient. We contribute to this literature by highlighting the role of $\lambda_1$ in the batch size scaling behavior of the model.

For this analysis, we focus on three of the WideResnet variants considered in Figure 2—the Batch-Norm model (a low curvature model), the non BatchNorm model (with moderate curvature), and the non BatchNorm model with 1.5X init scaling (with high curvature). We train these models while sweeping both the learning rate and the batch size.[4] We then measure the number of training steps required to reach $85\%$ validation accuracy, and the optimal learning rate found for each batch size. Similar to Shallue et al. (2018), we normalize the plotted steps to $85\%$ accuracy by the value measured at batch size $64$.

The results are shown in Figure 6. A few observations are in order: The low curvature model shows almost linear speedups in training speed as the batch size increases. In contrast, the high curvature model exhibits only minimal improvements in training speed with larger batch sizes. These scaling differences are closely mirrored by how the optimal learning rate $\eta^*$ changes with the batch size: for the low curvature model $\eta^*$ increases linearly with the batch size, while for the high curvature model $\eta^*$ is fixed around $3 \times 10^{-3}$. Notably, for the high curvature model $\eta^*$ is almost always the

---

[4]We sweep for the optimal learning rate on a log-scale grid between $10^{-3}$ and 1. For batch size, we sweep over powers of 2 from 16 to 4096.

largest non-divergent value—a clear indication that the high loss curvature slows down training by preventing larger values from being used.

A clear picture emerges from these observations. Previous research suggests that in order to effectively leverage larger batch sizes, one has to increase the learning rate in tandem with the batch size Jastrzębski et al. (2017); Goyal et al. (2017); Shallue et al. (2018); McCandlish et al. (2018). Our results suggest that large values of $\lambda_1$ place a sharp limit on the maximum the learning rate possible and therefore, limit the model's ability to leverage data parallelism effectively.

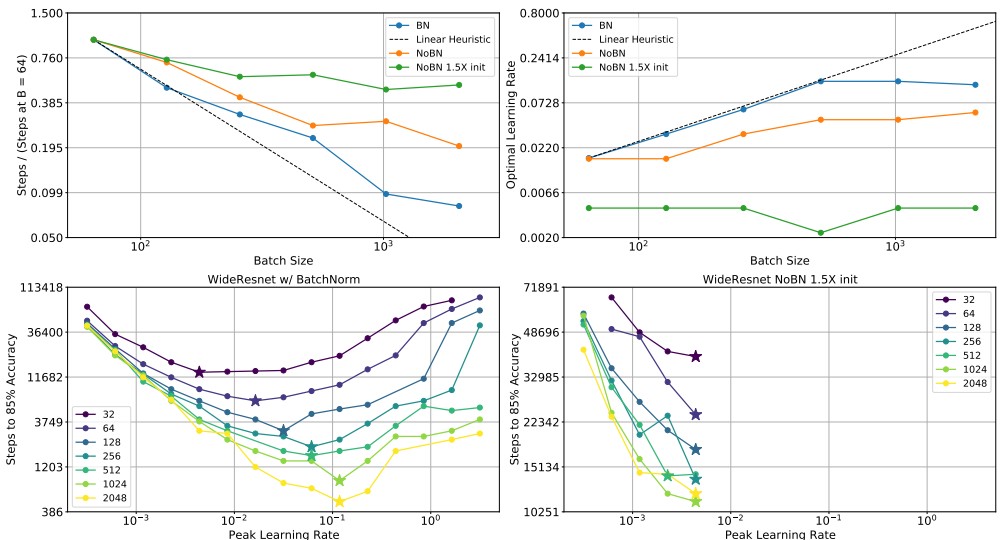

Figure 6: **Large curvature models scale poorly with batch size.** The four plots explore how the WideResnet models from Figure 2 scale with batch size. We look at three models, a low curvature model (WideResnet with BatchNorm) a medium curvature model (WideResnet without BatchNorm), and a high curvature model (WideResnet w/o BatchNorm and 1.5 init scaling). The low curvature model exhibits almost perfect (linear) scaling as the batch size increases, with the optimal learning rate increasing almost linearly with the batch size. The high curvature model shows almost no speedups at larger batch sizes, with the optimal learning rate fixed at the largest value with does not diverge. *Top left:* Steps required for each model to reach 85% accuracy, normalized by the steps required at batch size 64. *Top Right:* Optimal learning rate for each batch size. *Bottom Left:* Steps to 85% accuracy for each learning rate, broken down by the batch size for the BatchNorm model. *Bottom Right:* Steps to 85% accuracy for the non BatchNorm model with 1.5X init scaling.

## 7 CONCLUSION

Through extensive empirical experiments measuring the evolution of the loss sharpness during training, we have demonstrated how different methods such as initialization, learning rate warmup, and normalization all enable higher learning rates to be used (without causing divergence) by reducing $\lambda_1$ during training. It is noteworthy that two of the most popular models we investigated (the popular post-activation variant of the Resnet-50 and the standard WideResnet 28-10) did not benefit from learning rate warmup, and exhibited small values of $\lambda_1$ throughout training at the learning rates we considered. Thus researchers and practitioners who primarily work with well-optimized architectures might never notice a benefit from using warmup. However, even seemingly trivial modifications to a working architecture can easily result in large values of $\lambda_1$ and thus instability early in training—a naive response to such a situation would be to dramatically reduce the learning rate or, even worse, abandon the modification being investigated all together. We hope the perspective presented in this work can help future researchers better navigate such situations, either through investigating different initializations, applying warmup and gradient clipping, or changing the location of normalization layers in the model.

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

# A  LIMITATIONS

Our analysis has focused primarily on models trained with SGD with momentum. This decision was motivated to reduce additional confounds that arise when using adaptive preconditioning. Notably, it is unclear what the analogue of $\lambda_1 \leq 2/\eta$ should be for a model trained with Adam. In the appendix, we provide evidence that loss curvature adaption to the learning rate does occur even for Transformer models trained with Adam, and that learning rate warmup results in the similar effect of the optimization trajectory being "pushed" to flatter regions. However we leave a deeper analysis into this for future work.

Finally, while our experiments certainly raise questions about the efficacy better model initialization has on further accelerating training, our measurements has focused primarily on the (lack of) influence initialization has on $\lambda_1$ mid training. It is possible that better initializations could have lasting influence on the broader Hessian eigenspectrum (for example improving the ration $\lambda_k/\lambda_1$ for smaller eigenvalues $\lambda_k$) and that our analysis is missing such an effect.

# B  BRIEF REVIEW OF THE LOSS HESSIAN, EIGENVALUES AND QUADRATIC STABILITY BOUNDS

For completeness, we include a formal definition of the fundamental mathematical quantities discussed in the paper. We derive most of this discussion from the relevant chapters in Horn and Johnson (2012) and Boyd et al. (2004). We refer the reader to these sources for a more detailed discussion.

## B.1  THE HESSIAN MATRIX

The second derivative or Hessian matrix of the loss function $L(\cdot)$ at a point $\theta \in \mathbb{R}^n$ is denoted by $H(\theta) \in \mathbb{R}^{n \times n}$ where $\forall 1 \leq i, j \leq n$

$$H(\theta)_{i,j} = \frac{\partial^2 L(\theta)}{\partial \theta_i \partial \theta_j}. \tag{1}$$

Moreover, by Schwarz's theorem, if the second partial derivatives of $L(\cdot)$ are continuous at $\theta$, the matrix $H(\theta)$ is symmetric. This is a broad condition that holds for all the loss surfaces we examine in the main text (beyond a set of measure zero).

## B.2  EIGENVALUES

**Definition 1** *Let $A \in \mathbb{R}^{n \times n}$. If a scalar $\lambda$ and a nonzero vector $x$ satisfy*

$$Ax = \lambda x, \qquad \lambda \in \mathbb{C}, x \in \mathbb{C}^n, x \neq 0 \tag{2}$$

*then $\lambda$ is called an eigenvalue of A and $x$ is called an eigenvector of A associated with $\lambda$.*

If $A$ is a symmetric real matrix (such as the Hessian matrix), $A$ can be factored as

$$A = Q \Lambda Q^T, \tag{3}$$

where $Q \in \mathbb{R}^{n \times n}$ is an orthogonal matrix and $\Lambda = diag(\lambda_1, \ldots, \lambda_n) \in \mathbb{R}^{n \times n}$ is a real diagonal matrix. Here, $\{\lambda_i\}_{i=1}^n$ are all of the eigenvalues of $A$. We order $\lambda_i$ such that $\lambda_1 \geq \lambda_2 \cdots \geq \lambda_n$. In this ordering, $\lambda_1$ corresponds to the maximum eigenvalue of $A$ and $\lambda_n$ corresponds to its minimum eigenvalue.

The maximum and minimum eigenvalues of $A$ satisfy the following important properties:

$$\lambda_1 = \sup_{x \neq 0} \frac{x^T A x}{x^T x}, \qquad \lambda_n = \inf_{x \neq 0} \frac{x^T A x}{x^T x}. \tag{4}$$

In particular, for any $x \in \mathbb{R}^n$, we have

$$\lambda_n \|x\|_2^2 \leq x^T A x \leq \lambda_1 \|x\|_2^2.$$

### B.3 Stability of Gradient Descent for Quadratic Loss

Now that we have established the basics, let's derive the stability condition for GD applied to a quadratic loss function. Note that Wu et al. (2018) and Cohen et al. (2021) provide more general bounds for the stability of SGD-type optimization algorithms. Here, we state & derive the stability condition for GD for the sake of completeness.

Let

$$L(\theta) = \frac{1}{2}\theta^T H\theta,$$

where $H$ is a symmetric matrix with non-negative eigenvalues. Let's consider GD dynamics starting from a random point $\theta_0$. Under GD with a fixed step-size $\eta > 0$, we have

$$\theta_{t+1} = \theta_t - \eta\nabla L(\theta_t) = \theta_t - \eta H\theta_t = (I - \eta H)\theta_t.$$

Continuing this iteration to step 0 yields

$$\theta_t = (I - \eta H)^t\theta_0. \tag{5}$$

As $t \to \infty$, this iteration is stable iff[5] the eigenvalues of $(I - \eta H)$ have absolute magnitude bounded by one, which can be stated equivalently as

$$1 - \eta\lambda_1 \geq -1 \iff 2 \geq \lambda_1\eta$$
$$\iff \frac{2}{\eta} \geq \lambda_1,$$

which is the exact condition discussed and explored in the main text.

---

[5]As $\theta_0$ was randomly chosen, we assume it has a non-zero overlap with all eigenvectors.

## C    MISCELLANEOUS FIGURES

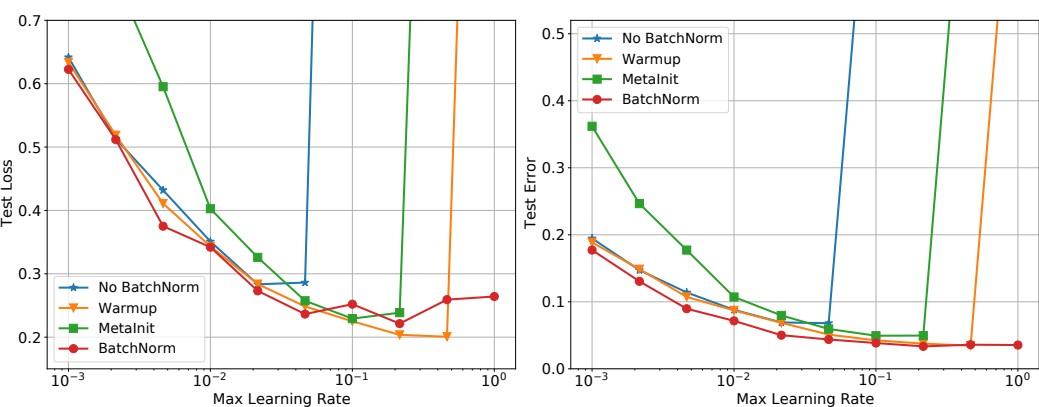

Figure 7: The test-time behavior of the different WideResnet models closely mirrors their training loss dynamics (Figure 1, left).

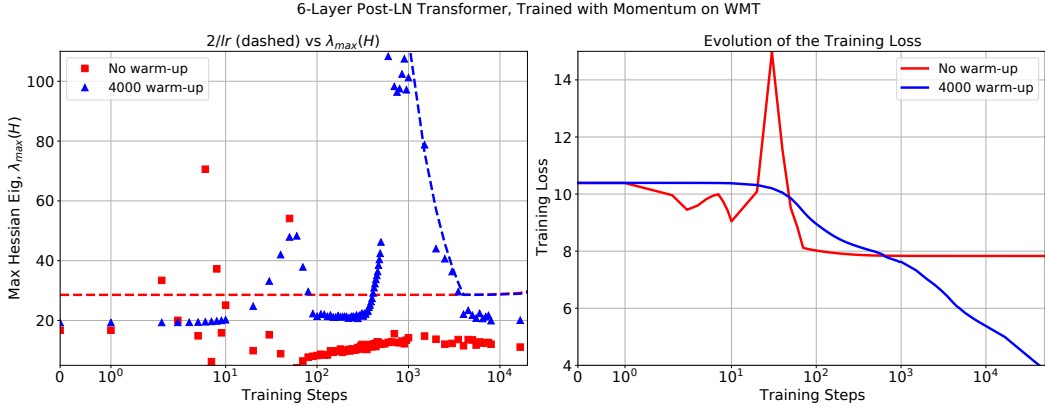

Figure 8: The curvature at initialization can be unreliable for predicting training stability. Left: Maximum eigenvalues of the Hessian throughout training. Dashed line correspond to $2/\eta$. Right: Evolution of the training loss for different models. The model with no warm-up (red) suffers from training instabilities even though it is below the stability threshold at initialization.

## D    PERFORMANCE OF MODELS IN FIGURE 2

In this this section, we plot the performance vs learning rate for all of the models shown in Figure 2 of the main text. These are shown in Figures 9, 13, 10, and 11. For models which diverged, we plot the best test performance achieved before divergence. In all settings, high curvature affects the final performance by limiting the use of higher learning rates.

We also noted several models in Figure 2 which diverged despite training starting out in a stable region of parameter space. In Figure 12 we plot the evolution of the loss sharpness during training, showing that it quickly enters a region where $\lambda_1 > 2.0/\eta$ before diverging around step 90.

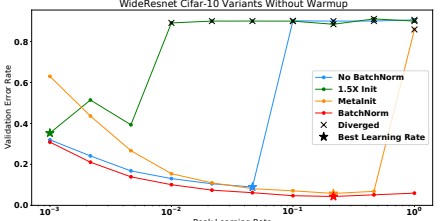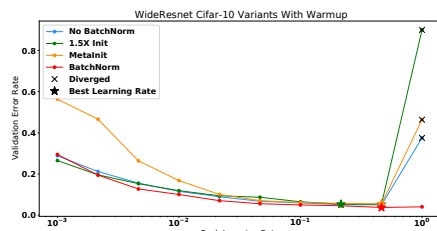

Figure 9: Performance of the WideResnet 28-10 models trained on Cifar10 shown in Figure 2 of the main text. (left) Performance of models trained without learning rate warmup. (right) Performance with learning rate warmup. When warmup is applied, all models reach comparable performance (though additional regularization is needed to match the generalization of the Batch Normalized models).

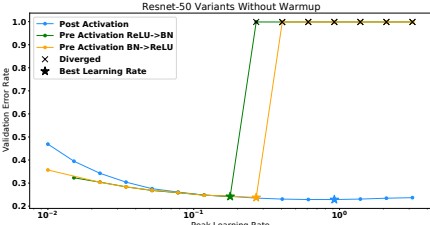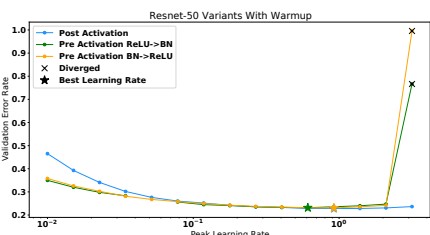

Figure 10: Performance of the Resnet50 models trained on Imagenet shown in Figure 2 of the main text. (left) Performance of models trained without learning rate warmup. (right) Performance with learning rate warmup.

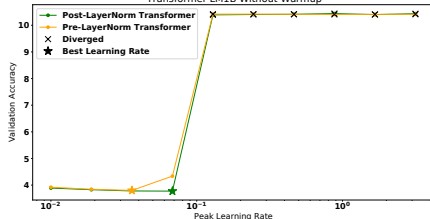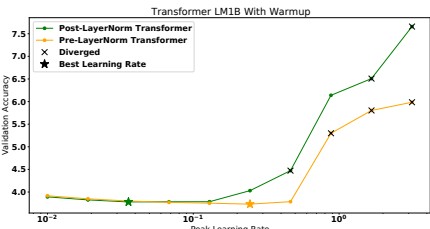

Figure 11: Performance of the Transformer models trained on LM1B shown in Figure 2 of the main text. (left) Performance of models trained without learning rate warmup. (right) Performance with learning rate warmup.

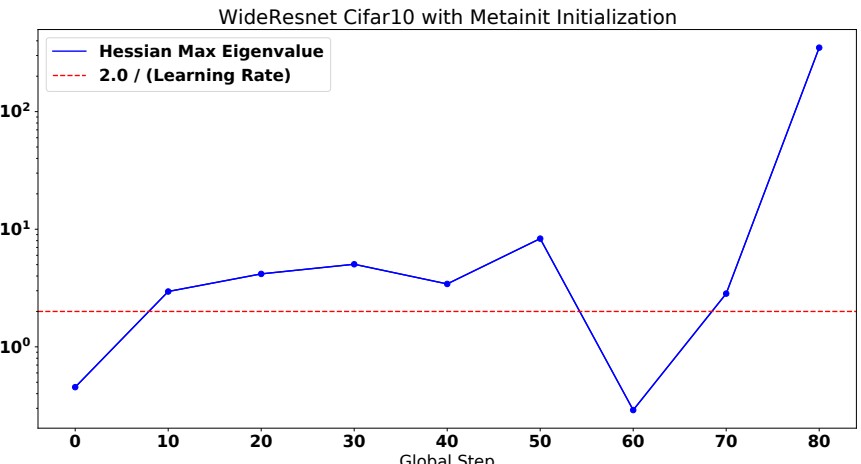

Figure 12: Training curve of the WideResnet 28-10 which diverged despite starting out in a region with $\lambda_1 < 2.0/\eta$. The parameters quickly leave the stable region and end up in a region of higher curvature before diverging.

### D.1 DISCUSSION OF STRIDE-(1,1) DENSENET EXPERIMENTS

In this section we discuss in more detail the Stride-(1,1) DenseNet experiments shown in Figure 2 in the main text. These experiments use a non-standard version of the DenseNet architecture where all average pooing strides are set to (1,1). Note the experiments in Figrue 5 and Table 1 instead use the standard strides implementation from the open sourced code of Zhu et al. (2021). The Stride-(1,1) DenseNet architecture is noteworthy because it is a counter example to common intuition that adding Batch Normalization results in flatter curvature. As shown in Figure 2 (left), the BN variants all have high curvature at initialization, however the right hand side plot shows that the mid training curvature becomes comparable to the non-BN variants. In Figure 13 we provide a more detailed analysis to understand what is happening with BN.

First we plot the performance of the BN vs non-BN models both with and without warmup. The differences are striking. Without warmup, we see the BN performance is highly stochastic, some trials outperform the non-BN variants, while some trials underperform the non-BN variants. However, when trained with 1000 steps of warmup the BN variants now significantly outperform the non-BN models are all considered learning rates. They can even be successfully trained at higher learning rates than the non-BN variants, despite the high initial curvature. To provide further detail, we show the training curves of select individual runs, both the evolution of the training loss and the evolution of curvature. The BN variants all exhibit catapult behavior early—the loss increases initially until the parameters enter a region of flatter curvature. Warmup helps the BN variants, and significantly reduces the severity of the catapult phase while enabling faster long term training. Additionally, when we add warmup we find that the BN variants can now be trained at higher learning rates than the non-BN variant. As shown at learning rate of .22, the non-BN model diverges during the warmup phase despite lower initial curvature.

Based on these experiments we arrive at the following conclusions. First, adding BN to the Stride-(1,1) DenseNet architecture results in high curvature at initialization, which results in a short period of instability during training. However, once the parameters escape this region of large curvature, the BN variant exhibits favorable training dynamics relative to the non-BN variants. Thus, there still seems to be benefit to adding BN, assuming steps are taken to mitigate the initial period of high curvature. The fact that adding BN to a model can result in high initial curvature is not without precedent, as Yang et al. (2019) observe that adding BN to deep fully connected networks can result in exploding gradients at initialization.

These experiments highlight one of the primary takeaways of this work: that maintaining flat curvature **throughout training** is a necessary (not sufficient) condition for stable training of neural networks. Thus it is not the presence of BN that is necessary for stable training, instead BN is generally a useful tool for reducing curvature (and thus stabilizing training). BN has clear benefits of improving curvature *in most cases*, but it is possible to produce configurations where adding BN paradoxically results in higher initial curvature than the non-BN variant. In these cases training is initially unstable, but once the curvature is reduced we see benefits later in training of using BN.

## E DETAILS ON COMPUTING THE HESSIAN EIGENSPECTRUM VIA LANCZOS

We use Lanczos iterations to estimate the top eigenvalue of the Hessian. Lanczos algorithm only requires Hessian-vector products which can be efficiently computed via Pearlmutter's trick Pearlmutter (1994). Previous research has demonstrated that this approach provides a robust and scalable framework to examine the eigenvalues of the Hessian for large neural networks Ghorbani et al. (2019); Papyan (2018).

For our WMT / LM1B experiments, we run the algorithm for $45$ steps while for image models, we use $40$ steps. When monitoring the evolution of the top eigenvalue as a function of the number of Lanczos steps, in all cases except one, we observe that the algorithm converges. For the case of Resnet with ReLU→BN ordering, due to a very small eigengap between the top eigenvalue and the bulk, the convergence is significantly slower. We use $200$ Lanczos steps in this case to alleviate the issue. **For this model, estimating $\lambda_1$ via power iteration (as is commonly done in the deep learning literature) will incorrectly the largest negative eigenvalue, not $\lambda_1$ as desired.**

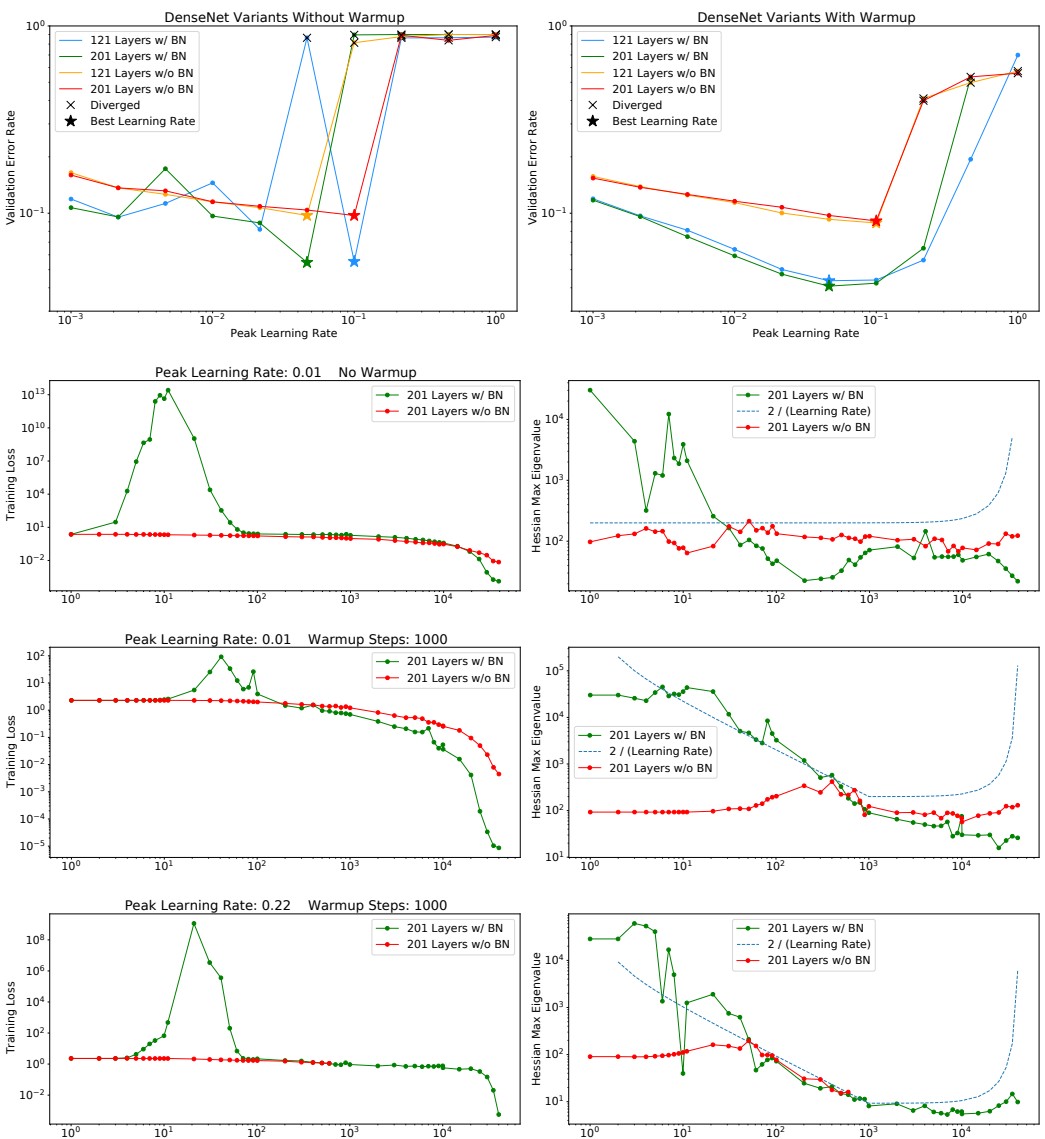

Figure 13: Performance of the Stride-(1,1) DenseNet models trained on Cifar10 shown in Figure 2 of the main text. **Top left:** Performance of models trained without learning rate warmup. The performance of BN has high stochasticity due to the large initial curvature, depending on how well the model recovers from the catapult phase. **Top Right:** Performance of models trained with 1000 steps of warmup. The BN variants now consistently outperform the non-BN variants, and can even be trained at higher learning rates despite the larger initial curvature. **Second Row:** Training curves and evolution of $\lambda_1$ for the BN and non BN model at learning rate .01 without warmup. The BN variant exhibits catapult behavior early but recovers once the curvature drops later in training. **Third Row:** Same as second row, but now with 1000 steps of warmup. The catapult behavior of the BN model is reduced, and training is faster after the catapult phase. **Bottom Row:** Comparing BN with non BN at a higher learning rate of .22 with 1000 steps of warmup. The BN model still has catapult behavior, but after recovering trains successfully once the curvature is reduced. The non-BN model diverges during the warmup phase just before step 1000.

It is well-known that Lanczos algorithm can suffer from numerical instabilities caused by finite-precision arithmetic. To alleviate these issues, Lanczos vectors are stored in float64 accuracy and we perform reorthogonalized at each step of the algorithm.

## F    Training Details for Table 1

### F.1    Neural Machine Translation

Neural Machine Translation experiments are based on the Transformer models (Vaswani et al., 2017). We use separate embeddings on encoder and decoder, and a common word piece vocabulary of size 32000. For depth, we use 6 layers on both encoder and decoder. For width, we experiment with two models, namely Transformer-Base and Transformer-Wide. For Transformer-Base, we use word embeddings with 512 dimensions, 8 heads and 2048 feed-forward dimension. For Transformer-Wide we use word embeddings with 1024 dimensions, 16 heads and 4096 feed-forward dimension. The experiments reported in Figures 3 and 5 use Transformer-Base. The experiments reported in Table 1 use Transformer-Wide models trained with Adam (Kingma and Ba, 2014). We sweep over warm-up, learning rate, gradient clipping and init_scaling and optimize for validation loss to evaluate performance on test set BLEU reported in Table 1. All the models are trained for 60 epochs at batch size of 1024 for Transformer-Base models, and batch size of 512 for Transformer-Big models. We use dropout of 0.1, label smoothing of 0.1 and no weight decay for all these models.

### F.2    DenseNets

In Table 1 the ResNet-50 (w/o BN) architecture was trained for 100 epochs at batch size 512, with l2 regularization of 5e-5, dropout of .3. It was trained with SGD with nesterov momentum of .9 and learning rate of .2. We applied gradient clipping at global l2 norm of 5 and used linear learning rate warmup with warmup period of 1000 steps.

For Table 1, the DenseNet-100 model was trained using the Gradinit codebase [6] by modifying the supplied DenseNet script to apply gradient clipping of norm 6 and to use the default initialization instead of GradInit.

## G    Training Details for Figure 2

The WideResnet-28-10 models were trained with batch size of 1024 for 300 epochs. We applied the MixUp augmentation(Zhang et al., 2017). For learning rate warmup we used 1000 steps of linear warmup until the peak learning rate is achieved, at which point the learning rate is decayed according to the cosine schedule.

The Stride-(1,1) DenseNet models were trained with batch size of 512 using the SGD optimizer with momentum of 0.9, weight decay of 5e-4, L2 regularization of 1e-4 and warmup of 1000 steps (for the models where warmup is used) followed by cosine decay. The models were trained for 200 epochs. For the DenseNet architecture we used growth_rate of 32 and reduction of 0.5.

The Resnet-50 models were trained with batch size of 2048 using the SGD optimizer with nesterov momentum of .9. The learning rate schedule was the same as in the WideResnet case, with linear warmup of 1000 steps followed by cosine decay. We applied label smoothing of .1 and used the standard flip plus crop for data augmentation.

The Transformer models on LM1B were trained at batch size 1024 using SGD with nesterov momentum of .9. We use embedding dimension of 512, 6 layers with 8 heads and MLP hidden dimension of 1024. The attention dropout rate was .1. The learning rate schedule followed the same recipe as in the Resnet cases.

## H    Curvature Adaptation with the Adam Optimizer

The discussion in the main text focused primarily on models trained with SGD and momentum. In this appendix, we briefly examine if similar conclusions hold for optimizers such as Adam that use preconditioning. It is unclear a priori whether or not curvature adapation to the learning rate should occur for optimizers which apply preconditioning. However, given that Adam is a diagonal preconditioner applied to a non-diagonal Hessian, there may be some similar effects observed.

---

[6]`https://github.com/zhuchen03/gradinit`

Consider a simple quadratic loss

$$L(\theta) = \frac{1}{2}\theta^T H\theta, \ H \succ 0.$$

where optimization is performed via preconditioned gradient descent with a fixed diagonal preconditioning matrix $D$:

$$\begin{aligned}
\theta_t &= \theta_{t-1} - \eta D^{-1}\nabla L(\theta_{t-1}) \\
&= \theta_{t-1} - \eta D^{-1}\big(H\theta_{t-1}\big) \\
&= \big(I - \eta D^{-1}H\big)\theta_{t-1} \\
&= \big(I - \eta D^{-1}H\big)^t\theta_0
\end{aligned}$$

As such, this simple model would suggest that the max eigenvalue of the following matrix may be related to training instability of models trained with Adam

$$\lambda_{max}(D^{-1}H) = \lambda_{max}(D^{-1/2}HD^{-1/2}). \tag{6}$$

While (6) does not take into the account the effects of adaptive preconditioning or momentum, we find some empirical evidence that this approximation provides understanding into the stability of the optimization.

Figure 14 below examines the evolution of $\lambda_{max}(D^{-1/2}HD^{-1/2})$ for three Transformer models trained with Adam and different warm-up lengths. Here, $D$ is a diagonal matrix with $D_{i,i} = \sqrt{\text{Corrected Adam grad squared EMA}} + \text{Adam } \epsilon$. We observe that, similar to the models trained with momentum, the maximum (preconditioned) Hessian eigenvalue adapts to the warm-up schedule (green and red markers). We notice that –perhaps due to the effect of momentum or adaptive preconditioning – the threshold $2/\eta$ does not seem to be aligning with the data well. Instead, an empirically corrected threshold $40/\eta$ seems to fit the data better. We observe that instabilities in model training coincide exactly with $\lambda_{max}(D^{-1/2}HD^{-1/2})$ crossing the empirically corrected threshold.

These observations suggest that some of the insights discussed in the main text seem to carry over to the case of adaptive optimizers. We leave further exploration of this more complex setting to future work.

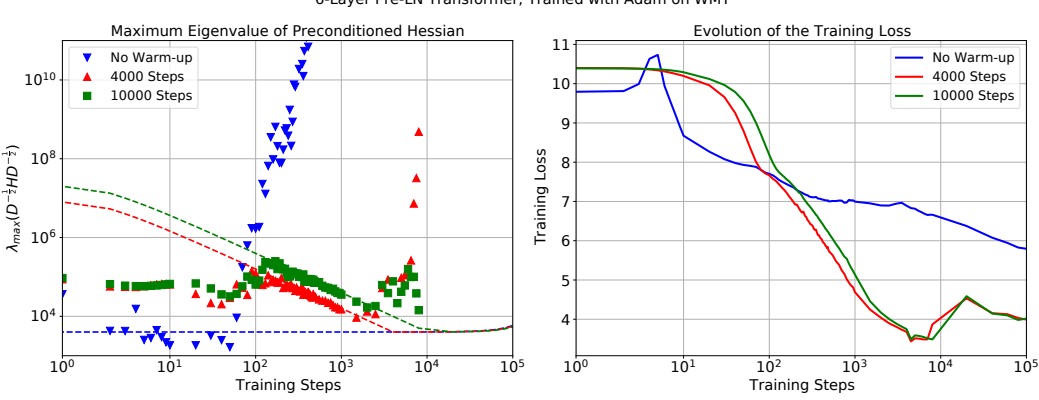

Figure 14: Instabilities in model training are reflected in the loss curvature even for models trained with Adam. Left: Maximum eigenvalues of the preconditioned Hessian throughout the training. Dashed line correspond to $40/\eta$. Right: Evolution of the training loss for different models. Training becomes unstable exactly when the eigenvalues cross the threshold.

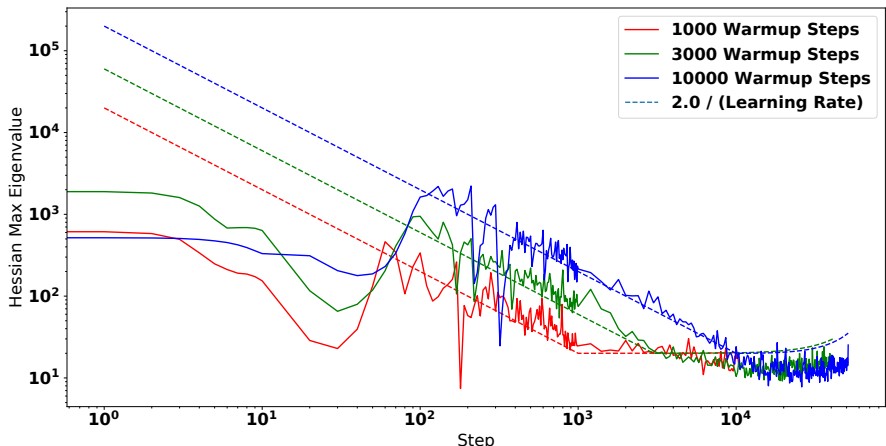

Figure 15: Further evidence of learning rate warmup "pushing" the optimization trajectory towards regions with reduced $\lambda_1$. Note the rate at which $\lambda_1$ changes closely matches depends on the length of the warmup. Model shown is the non-BN WideResnet (standard init) trained at batch size 2048 with peak learning rate of .1.

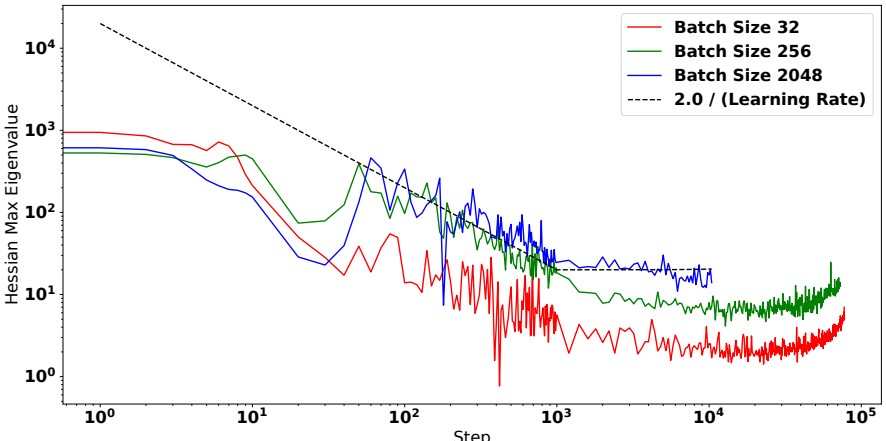

Figure 16: Wu et al. (2018) predicts that the stability bound will become tighter than $2/\eta$ at smaller batch sizes. This plot confirms holds not only at convergence (as implied by their Theorem 1), but early in training as well. For a small batch size (32), the mid training curvature hovers significantly below the $2.0/\eta$ approximation. As the batch size increases to 2048, the mid training curvature is larger, approaching the $2.0/\eta$ approximation. All curves show the WRN without batch norm, trained with the same learning schedule using 1000 steps of warmup.

## I COMPUTE RESOURCES USED

Nearly all experiments utilized the Google Cloud Platform with v2 cloud TPUs except for the following: The Figure 2 Resnet-50 and Stride-(1,1) Densenet experiments utilized the v3 cloud TPU, while the GradInit code was run on a cloud machine with a single V100 GPU. The Figure 2 experiments were done in parallel using up to 50 v2 TPUs concurrently over the period of a few days. Additionally, all the Machine Translation models were trained on v3 cloud TPUs.

