# OpenReview forum: "A Loss Curvature Perspective on Training Instabilities of Deep Learning Models"
_ICLR.cc/2022/Conference — ICLR 2022 Poster_

### Official Review · Reviewer_P6at · 2021-11-02

**Correctness:** 4
**Technical Novelty And Significance:** 3
**Empirical Novelty And Significance:** 3
**Recommendation:** 6
**Confidence:** 4

**Main Review:**

Comments :
- Results are well presented and yield a rather coherent picture. In particular, I like the intuition that warmup brings to a region of smaller curvature, which fits in well with the catapult mechanism. I also appreciate the batch size scaling experiments, which go against the idea that the only quantity that matters is the ratio of learning rate / batch size.
- The observation that lambda_1 follows 2/lr in presence of warmup could be better connected to Cohen et al. with a sentence such as “Cohen et al. showed that lambda_1 increases up to 2/lr then plateaus at the edge of stability, in the constant lr setup : we show that this holds even in presence of scheduling”
- The figures are extremely long to process for the various PDF readers I tried. This is likely due to an excessive number of datapoints in the logscale figures (happened to me in the past) : consider subsampling the data at large values.

Questions :
- Fig. 3 : Could one try to understand why some of the models diverge even though eta<2/lambda_1 ?
- Fig. 3 : Multiplying the init variance by 1.5 seems to increase the sharpness by a factor between 5 and 100. Why such a variability ? Can one predict this factor from the number of layers in the model ?
- If increasing init variance increases sharpness, couldn’t one simply reduce the sharpness by reducing init variance, hence enabling larger learning rates ? This could be connected to the question of lazy vs feature learning regimes

**Summary Of The Paper:**

This paper presents a comprehensive set of experiments showing how the maximal learning rate to train modern architectures depends on various algorithmic choices.

**Summary Of The Review:**

Although the results presented are not particularly groundbreaking, the paper is rather well written and easy to follow, and gives some potentially useful insights on an important topic : learning rate scheduling. Therefore, I think this paper could be of interest to the community.

---

### Official Review · Reviewer_vdgP · 2021-11-02

**Correctness:** 4
**Technical Novelty And Significance:** 2
**Empirical Novelty And Significance:** 3
**Recommendation:** 8
**Confidence:** 3

**Main Review:**

The paper seems well written and manages I think to condense a significant amount of empirical work in the short limit we have. I think the analysis that has been done across multiple architectures and models provides a good basis for the authors claims in the text.

Strengths:
* The paper contains a very solid set of large scale and in-depth evaluation, which is something well needed for this kind of empirical work.
* The different analysis done well presents the vast amount of results and the key takeaways from the experiments

Weaknesses:
 * Since we do observe models that diverge despite beginning in stable regions (Fig. 3 D) it seems that there the initial warnup has a significant effect of where it takes the model beyond its initialization, which is not too well discussed.
 * In the Dense Net experiment in Table 1, there is no reported results for including warmup




**Summary Of The Paper:**

The paper performs extensive empirical investigation into factors that play important role in making neural network trainable. One of the key quantity that the authors find to give significant insights is the largest eigenvalue of the Hessian matrix. They show that models that train successfully tend to have learning rates close to the well known critical bound for GD - 2 / \lambda. Further investigation to various architectural choices - such as normalizations and initialization techniques indicate that the value of \lambda behave somewhat consistently after the initial training period for well training models. In addition, they demonstrate that learning rate wramup can provide a significant improvement by pushing parameters in regions with lower values of \lambda.

**Summary Of The Review:**

Strong empirical paper which well presents many experiments and observations of the interplay of the maximum eigenvalue of the Hessian and learning dynamics.

---

### Official Review · Reviewer_RGH4 · 2021-11-03

**Correctness:** 4
**Technical Novelty And Significance:** 1
**Empirical Novelty And Significance:** 4
**Recommendation:** 8
**Confidence:** 4

**Main Review:**

Strengths:

* The paper includes quite a lot of experimental results and definitely provides valuable empirical observations (see them in the brief summary above). It gives an insightful and convincing understanding of the learning rate warmup strategy. It also points out several failure cases where good curvature does not guarantee convergence in training, which is equally valuable.

Here are the questions for each section:

* Section 4.
I see slightly different max eigenvalue at initialization for same architecture with different learning rate -- is it fully due to the randomness in the initialization? Or do I miss something here?

* Section 5.
Explain MetaLoss briefly.

* Section 6.
  1. Given Figure 3, why would the authors not include MetaInit as another baseline to test batch size scaling?
  2. In Figure 6, what learning rate is used for the Top left plot?
 I'd guess it's the optimal (peak) learning rate per batch size, but it's not fully clear. If not, please clarify and better help me understand how the learning rate is chosen; If yes, it looks like BN with batch size 128, 256, 512 has a log-linear improvement from the top left plot, but not very log-linear if looking at the bottom left plot: improvement from 128 to 256 is much larger than the improvement from 256 to 512. Similar results for NoBN 1.5x init. So I may be missing something here.

**Summary Of The Paper:**

The paper explores training instabilities of deep learning models by monitoring the max eigenvalue of the Hessian in i) different architectures; ii) different optimization tricks; iii) different stages of training.

The paper is a mix of several empirical observations:

(Section 4) One of the main observations is that the necessary (not sufficient) requirement for a "successful" training is to have *relatively* low max eigenvalue through the training process, where *relative* is roughly determined by the inverse of the learning rate.

(Section 5) Second main observation is specific to the learning rate warmup strategy: the warmup strategy pushes max eigenvalue to the boundary below $2/\eta$, and with a better understanding of this optimization strategy, the authors show comparable performance in multiple datasets compared with other optimization tricks.

(Section 6) Lastly, the authors explore batch size scaling with different scales of max eigenvalue, focusing on comparing BN (with small curvature only for that specific task), NoBN (larger curvature), NoBN 1.5xinit (highest curvature). The experiment shows for small curvature setting, both batch size and learning rate can scale quite well.

**Summary Of The Review:**

Overall, this is a good empirical paper with lots of experimental results and good insights. The learning rate warmup part is the most insightful among all.

---

### Official Review · Reviewer_oiwU · 2021-11-09

**Correctness:** 3
**Technical Novelty And Significance:** 2
**Empirical Novelty And Significance:** 2
**Recommendation:** 5
**Confidence:** 4

**Details Of Ethics Concerns:**

The authors present investigations regarding the connections between curvature and many initialization methods and techniques. However, no clear conclusions are made and some investigations are not deep enough.


**Main Review:**

The paper presents the necessity of low curvature at the early stage of training for stable neural network training with large learning rate, some notable observations includes “the mid-training conditioning is determined by the learning rate, not on the initialization method used”; “Learning rate warmup can match the performance of recent advances in initialization research”. It seems to suggest that the weight initialization does not matter that much and warmup can be mitigated.

However, the paper is kind of hard to read as there are too many empirical observations presented but not quite well organized. For example, the most measurement \lamda_1 is is not clearly introduced and there is only a small footprint saying lambda_1 refers to maximum eigenvalue of the loss Hessian. The reason why 2/\eta is chosen is also not clear.

The curvature and sharpness: the authors use the term sharpness to denote the max eigenvalues of loss Hessian to denote the curvature or the the loss surface, however, there are other sharpness/smoothness measurements such as [1, 2], which were used to denote the generalization ability of the loss surface at the end of training. It is easy to get confused with those terms for different conditions. Are there any connections? If so, why not study them as well? It would be great to know how those metrics are correlated with the curvature. If not, it would be better to make clear the difference and not use the terms interchangeably.


The authors have shown that many techniques actually made low curvature. However, curvature itself can not reliably determine the trainability. As noted by the authors,  some models can be successfully trained even when they start out in the unstable region and measuring \lamda_1 at initialization is not always sufficient to predict whether or not the model will be easily trained.

On the other hand, the DenseNet experiments says that the models with Batch Normalization actually start out with higher curvature than the non-BN variants, which suggests that curvature may not be able to explain the effects of BN quite well and no smoothness benefits are observed at initialization. I would hope there are more investigations here as BN is still a critical module and the authors observation contradicts previous belief.


[1] Keskar et al, On large-batch training for deep learning: Generalization gap and sharp minima. ICLR 2017
[1] Jiang et al, Fantastic Generalization Measures and Where to Find Them, ICLR 2020

**Summary Of The Paper:**

The paper aims to understand what makes general architectures trainable, specifically what limits the maximum learning rate for deep learning models trained with SGDM, from the loss curvature aspective. They empirically studied the evolution of the loss sharpness as they vary the learning rate, warmup period, initialization, and architectural choices. They show maintaining a sufficiently small \lambda_1 is a necessary condition for successful training at a large learning rate. And different methods such as initialization, learning rate warmup, and normalization all enable higher learning rates to be used by reducing \lambda_1 during training. More specifically:
- Some initialization strategies for architectures without normalization actually operate by reducing curvature early in training, enabling training at larger learning rates.
- learning rate warmup gradually reduces \lambda_1 during training, which is a competitive baseline in comparison with better model initialization methods.
- large loss curvature can result in poor scaling at large batch sizes and interventions designed to improve loss conditioning can improve the model’s ability to leverage data parallelism.

**Summary Of The Review:**

The authors present investigations regarding the connections between curvature and many initialization methods and techniques. However, no clear conclusions are made and some investigations are not deep enough.

---

> ### Author Response · Authors · 2021-11-16
> **Response Part 1**
>
> Thank you for the time reviewing our paper. We have uploaded a revised version of the paper based on your comments which we believe will improve clarity. Please let us know if you have additional questions and concerns, we are happy to engage in discussion in order to improve the paper.
>
>
> "On the other hand, the DenseNet experiments says that the models with Batch Normalization actually start out with higher curvature than the non-BN variants, which suggests that curvature may not be able to explain the effects of BN quite well and no smoothness benefits are observed at initialization. I would hope there are more investigations here as BN is still a critical module and the authors observation contradicts previous belief."
>
>
> Based on your recommendation we have added Section D.1 to the Appendix and a new Figure 13 to further explore the DenseNet experiments. Indeed, as we will discuss, there are some subtleties to the DenseNet experiments that Figure 2 doesn't capture. Please take a look at Figure 13, which shows the full time evolution of the loss curvature for select DenseNet+BN experiments. A few observations are in order:
>
> 1. The DenseNet+BN models have significantly higher curvature at initialization than the nonBN variants. Correspondingly, for modest learning rates these models have the loss actually increase early in training, while the non BN variants have stable training throughout. There is prior work which has observed similar behavior can occur with BN, as [1] show that deep MLPs with BN have exploding gradients at initialization.
> 2. The DenseNet+BN models eventually recover from the high initial loss curvature, and begin to train stably after the curvature has decreased (this is akin to the catapult mechanism from [2]).
> 3. When combined with warmup, the DenseNet+BN variants actually outperform the non-BN variants. Suggesting that once the large curvature at initialization has been reduced, BN improves training as is more aligned with conventional wisdom.
>
> Thus the DenseNet experiments actually strengthen the main conclusions of the paper in the following ways:
>
> 1. Maintaining flat curvature throughout training is a necessary (not sufficient!) condition for stable optimization (DenseNet+BN models unstable early when curvature is high, but become stable later when the curvature falls). Note, this experiment shows that it is the loss curvature that matters for stability, not necessarily the presence of BN. BN only helps when it actually reduces curvature (it usually does, but with exceptions).
> 2. Curvature at initialization is only loosely correlated with stable training. The loss curvature is a dynamic quantity that evolves throughout training, it can start high and then shrink (early instability followed by stable training late), or it can start small then grow (early stability followed by late instability).
> 3. Warmup is an effective way to mitigate large curvature early in training.
>
>
> [1] Yang et. al. A mean field theory of batch normalization
>
> [2] Aitor Lewkowycz et. al. The large learning rate phase of deep learning: the catapult mechanism

---

> ### Author Response · Authors · 2021-11-16
> **Response Part 2**
>
>
> "For example, the most measurement \lamda_1 is is not clearly introduced and there is only a small footprint saying lambda_1 refers to maximum eigenvalue of the loss Hessian."
>
> We agree it would be beneficial to provide a precise definition of \lambda_1, however given that we cite substantial prior work looking at the loss curvature of deep learning models we left it out of the main text. We have added a new section to the appendix (Appendix B) that defines this properly.
>
> "The reason why 2/\eta is chosen is also not clear."
>
> We have added Section B to the Appendix to include a more detailed discussion, though we discuss this briefly in the introduction:
> "and quadratic models of the loss surface predict that optimization with SGD is unstable when λ1 > 2/η (Wu et al., 2018).". Also in the related works we discuss several prior works noting that deep learning training occurs near this bound. To summarize existing knowledge, optimization of a quadratic loss with SGD provably diverges when λ1 > 2/η. Deep learning models of course aren't quadratic, however empirically we demonstrate that maintaining small curvature throughout training is a necessary condition for successful training with SGD.
>
>
> "The curvature and sharpness: the authors use the term sharpness to denote the max eigenvalues of loss Hessian to denote the curvature or the the loss surface, however, there are other sharpness/smoothness measurements such as [1, 2], which were used to denote the generalization ability of the loss surface at the end of training. It is easy to get confused with those terms for different conditions. "
>
> Note that Keskar et. al. define loss sharpness the same way we do (magnitude of the largest Hessian eigenvalue), however they use an approximation of this quantity in the experiments for computational reasons. Quoting directly from their Section 2.2.2: "Sharpness of a minimizer can be characterized by the magnitude of the eigenvalues of ∇2f(x), but given the prohibitive cost of this computation in deep learning applications, we employ a sensitivity measure that, although imperfect, is computationally feasible, even for large networks.". However, since this work there has been substantial progress on efficient and exact computation of the outlier eigenvalues of the Hessian (see Ghorbani et. al.) which is why we no longer need approximations. Given that we don't need approximations of the loss sharpness, we didn't feel it necessary to test how correlated the Keskar et. al. approximation is with the true value.
>
>
> "The authors have shown that many techniques actually made low curvature. However, curvature itself can not reliably determine the trainability. As noted by the authors, some models can be successfully trained even when they start out in the unstable region and measuring \lamda_1 at initialization is not always sufficient to predict whether or not the model will be easily trained."
>
>
> As noted by Reviewer RGH4, the main takeaway of Section 4 is that "the necessary (not sufficient) requirement for a "successful" training is to have relatively low max eigenvalue through the training process, where relative is roughly determined by the inverse of the learning rate.". One consequence of this is that curvature at initialization is only loosely correlated with successful training, however as is clearly shown in the right hand side plots of Figure 2, all non-divergent models in mid training enter a region where lambda_1 < 2 / lr. Figure's 3 and 4 shows that these successful models do indeed maintain this bound throughout training. Crucially, this is only a necessary condition, as we provide several examples where models start out in the stable region, but diverge anyway as the curvature can grow mid training. This observation has implications for our understanding of NN training, as many papers (e.g. most of the NTK literature) focus on model conditioning at initialization, this type of analysis can be unreliable given how rapidly conditioning evolves during training.

---

### Decision · Program_Chairs · 2022-01-20

**Decision:**

Accept (Poster)

**Comment:**

The paper provides a thorough study of the evolution of Hessian depending on a wide variety of aspects such as initialization, architectural choices, and common training heuristics. The paper makes a number of interesting observations. Some of them are not really new but overall, the experimental evaluation of the paper makes it a valuable resource for the community.

The reviewers are overall quite positive. One reviewer notes that more investigation of the behavior of batch-normalization is required. I encourage the author to address this concern in the final manuscript. There is a lot of recent work on batch-normalization that might be worth discussing, e.g.:
Training BatchNorm and Only BatchNorm: On the Expressive Power of Random Features in CNNs
Jonathan Frankle, David J. Schwab, Ari S. Morcos

Batch Normalization Provably Avoids Rank Collapse for Randomly Initialised Deep Networks
Hadi Daneshmand, Jonas Kohler, Francis Bach, Thomas Hofmann, Aurelien Lucchi

A Quantitative Analysis of the Effect of Batch Normalization on Gradient Descent
Yongqiang Cai, Qianxiao Li, Zuowei Shen